# Being noisy in a crowd: Differential selective pressure on gene expression noise in model gene regulatory networks

Nataša Puzović[1]*, Tanvi Madaan[1], Julien Y. Dutheil[1,2]

**1** Molecular Systems Evolution Research Group, Max Planck Institute for Evolutionary Biology, Plön, Schleswig-Holstein, Germany, **2** Institut des sciences de l'évolution, Montpellier, Languedoc-Roussillon, France

\* puzovic@evolbio.mpg.de

## Abstract

Expression noise, the variability of the amount of gene product among isogenic cells grown in identical conditions, originates from the inherent stochasticity of diffusion and binding of the molecular players involved in transcription and translation. It has been shown that expression noise is an evolvable trait and that central genes exhibit less noise than peripheral genes in gene networks. A possible explanation for this pattern is increased selective pressure on central genes since they propagate their noise to downstream targets, leading to noise amplification. To test this hypothesis, we developed a new gene regulatory network model with inheritable stochastic gene expression and simulated the evolution of gene-specific expression noise under constraint at the network level. Stabilizing selection was imposed on the expression level of all genes in the network and rounds of mutation, selection, replication and recombination were performed. We observed that local network features affect both the probability to respond to selection, and the strength of the selective pressure acting on individual genes. In particular, the reduction of gene-specific expression noise as a response to stabilizing selection on the gene expression level is higher in genes with higher centrality metrics. Furthermore, global topological structures such as network diameter, centralization and average degree affect the average expression variance and average selective pressure acting on constituent genes. Our results demonstrate that selection at the network level leads to differential selective pressure at the gene level, and local and global network characteristics are an essential component of gene-specific expression noise evolution.

## Author summary

"No man is an island, entire of itself. Each is a piece of the continent, a part of the main." declares John Donne in his poem *For Whom the Bell Tolls*, emphasizing that no individual human is entirely separate from humanity as a whole interconnected system. Organisms are biological systems constituted of many interacting components that also interact with each other and the environment. Understanding the evolution of single components such

**Data Availability Statement:** The simulation results data and the code necessary to reproduce all figures is available at https://doi.org/10.5281/

zenodo.6939845, together with the code necessary to generate all raw simulation files.

**Funding:** NP is funded by the International Max Planck Research School (IMPRS) for Evolutionary Biology. The funders had no role in study design, data collection and analysis, decision to publish, or preparation of the manuscript.

**Competing interests:** The authors have declared that no competing interests exist.

as individual cells or genes can only be fully achieved by considering the interactions with other components. Here, we study the evolution of the cell-to-cell variability of gene expression, the so-called expression noise. To understand the evolution of gene-specific expression noise, we develop a model of gene network evolution with selection at the gene regulatory network level. We find that selection at the gene network level has different repercussions for individual genes based on their position in the network and that gene expression noise is more constrained in genes that are central in the network. Furthermore, the topological structure of the background network affects the propagation and evolvability of gene expression noise. These findings indicate that selection on a given system results in differential selective pressures at the level of subsystems. Our results further suggest that selection to mitigate inherent noise plays a role in network and gene evolution.

## Introduction

Living beings are complex systems constituted of many genes that interact with each other and the environment to create an organism. From prokaryotes with a few hundred essential genes, to eukaryotes with possibly several thousands, cells require many gene products to work together to perform housekeeping functions and to replicate. Fine-tuned molecular processes, generally referred to as *gene expression*, ensure how, where and when these products are generated. However, gene expression is an inherently noisy process [1, 2], which involves many steps where molecules participating in the expression machinery diffuse and bind to target molecules. Additionally, these molecules are often present in small copy numbers, increasing the susceptibility of gene expression to stochastic events. Consequently, there is a variation in gene expression levels among cells, even if they are isogenic and grown in a homogeneous environment, and this inevitable variation has been termed *gene expression noise*. Organisms have to express hundreds of genes, each one of which is noisy—raising the question of how they evolved to cope with this inevitable noise.

The expression noise level of a particular gene may be decomposed into two components, called *extrinsic* and *intrinsic*. Extrinsic noise affects all genes equally and results from the sharing of key molecules, such as RNA polymerases and ribosomes, by all genes in the expression process, as well as, for instance, differences in cell size and phase in the cell cycle. Intrinsic noise is gene-specific and results from different chromatin states, cis-regulatory elements and kinetic parameters of transcription and translation of each gene [3]. Minor sequence mutations can have a significant effect on the level of expression noise. For example, a small number of single-nucleotide changes in a transcription factor binding site were reported to have a large effect on the expression noise level [4]. Since (i) there is variation in the level of intrinsic noise of genes, and (ii) intrinsic noise is genetically determined—and, therefore, heritable—gene expression noise can be shaped by natural selection.

Evidence of selection on expression noise was first seen in the fact that dosage-sensitive genes [5] and essential genes exhibit lower levels of expression noise [6, 7]. Intrinsic noise was also reported to correlate with the strength of selection acting on the encoded protein. Namely, proteins with a lower ratio of non-synonymous over synonymous substitution rate (Ka/Ks) have a lower level of expression noise [8]. Changes in the expression noise of a single gene may be either beneficial or deleterious, depending on how far its mean expression is from the optimal expression level [9]. Expression noise is deleterious if the mean expression level is close to the optimal, as higher variation, in this case, generates a larger number of less fit individuals,

reducing the population fitness. Conversely, expression noise can be beneficial if the mean expression level is far from the optimum, as noisy genes are more likely to generate cells with an expression level closer to the optimum. Noisy gene expression can thus be part of a bet-hedging strategy and was observed in genes involved in immune and environmental response [10–13]. The fitness cost of changes in the level of expression noise in the fitness landscapes of ≈ 30 yeast genes have been shown to be on the same order as fitness costs of changes in mean expression level [14]. Since the fitness effect of different levels of expression noise can be as detrimental as different mean expression levels, which are thought to be extensively under selection [15], it can be assumed that expression noise is extensively under selection genome-wide. Prevalent selection on expression noise has been demonstrated in naturally segregating promoter variants of *E. coli* [16].

The phenotype (and, therefore, the fitness) of an organism depends on the interaction of many genes. As a result, genes do not evolve independently, and the selective pressure acting on a gene's intrinsic noise depends on its interactions with other genes. Understanding the evolution of gene expression noise requires accounting for such gene-to-gene interactions, commonly depicted by a gene network. The propagation of noise from gene to gene in the network was established both theoretically and experimentally [17, 18]. Genes with many connections propagate their noise to a more substantial extent than genes with fewer connections and, therefore, contribute more to the global noise levels of the network. Gene networks are robust to variation in the expression level of their system components to some degree, but at a critical point the global noise of the network becomes too high and leads to network collapse. Selection against noise at the network level was, therefore, hypothesized to result in stronger constraints on the intrinsic noise of highly connected genes [8]. Moreover, the topological structure of the network has been shown to affect the pattern of noise propagation [19], suggesting that the topology of the network might impose additional selective constraints on the constituent genes.

Here, we test the hypothesis that expression noise of highly connected genes in gene networks is under stronger selective pressure than expression noise in peripheral genes using an *in silico* evolutionary experiment. We introduce a new gene regulatory network evolution model, which includes an evolvable component of stochastic gene expression, and use it to evolve thousands of network topology samples over 10,000 generations. These simulations showed that highly connected genes have a more constrained intrinsic expression noise. They further revealed that not all genes might evolve in response to network-level selection, and the probability that they do so depends on local network properties. Lastly, the average selective pressure acting on genes in a network is affected by topological features such as network diameter, centralization and average degree.

## Materials and methods

We introduce a new gene regulatory network model that incorporates intrinsic expression noise. We then use this model within a forward simulation framework to simulate the evolution of populations of networks with mutable levels of intrinsic expression noise. These simulations allow us to study how the selective pressure acting on expression noise varies within the regulatory network.

### A gene regulatory network model with stochastic gene expression

To investigate the evolution of stochastic gene expression in gene regulatory networks, we first extend Wagner's gene network model [20] to integrate gene-specific expression noise.

We model a network of $n$ genes ($n = 40$ in this study) defined by a regulatory matrix $W = (w_{ij})_{1 \leq i \leq n, \, 1 \leq j \leq n}$, and a vector of intrinsic, gene-specific noise $\{\eta_i^{\text{int}}\}_{1 \leq i \leq n}$. Each element $w_{ij}$ of

the regulatory matrix $W$ defines the regulatory effect of gene $j$ on gene $i$. The value of $w_{ij}$ is a real number and is referred to as regulatory strength of gene $j$ on gene $i$. In case $w_{ij} > 0$, gene $j$ is an activator of gene $i$ and increases its expression level. Conversely, when $w_{ij} < 0$, gene $j$ is a repressor of gene $i$ and decreases its expression level. Lastly, if $w_{ij} = 0$, gene $i$ is not regulated by gene $j$ and gene $j$ has no effect on expression level of gene $i$. Two genes $i$ and $j$ are connected by an edge in the network if at least one of $w_{ij}$ and $w_{ji}$ is non-null. The intrinsic noise vector $\{\eta_i^{\text{int}}\}_{1 \leq i \leq n}$ defines the gene-specific expression noise of each gene in the network. The regulatory matrix and the intrinsic noise vector together constitute a unique genotype in this modeling framework (Fig 1A).

The phenotype (the expression level of each gene) in the model is represented by a state vector $\{S_i\}_{1 \leq i \leq n} = \{s_1, s_2, \ldots, s_n\}$, which describes the expression level of each gene. The state vector at $t_0$ is set to an arbitrary basal expression level value ($\{S_i^0\}_{1 \leq i \leq n} = \{S_i^{basal}\}_{1 \leq i \leq n} = \{20, \ldots, 20\}$

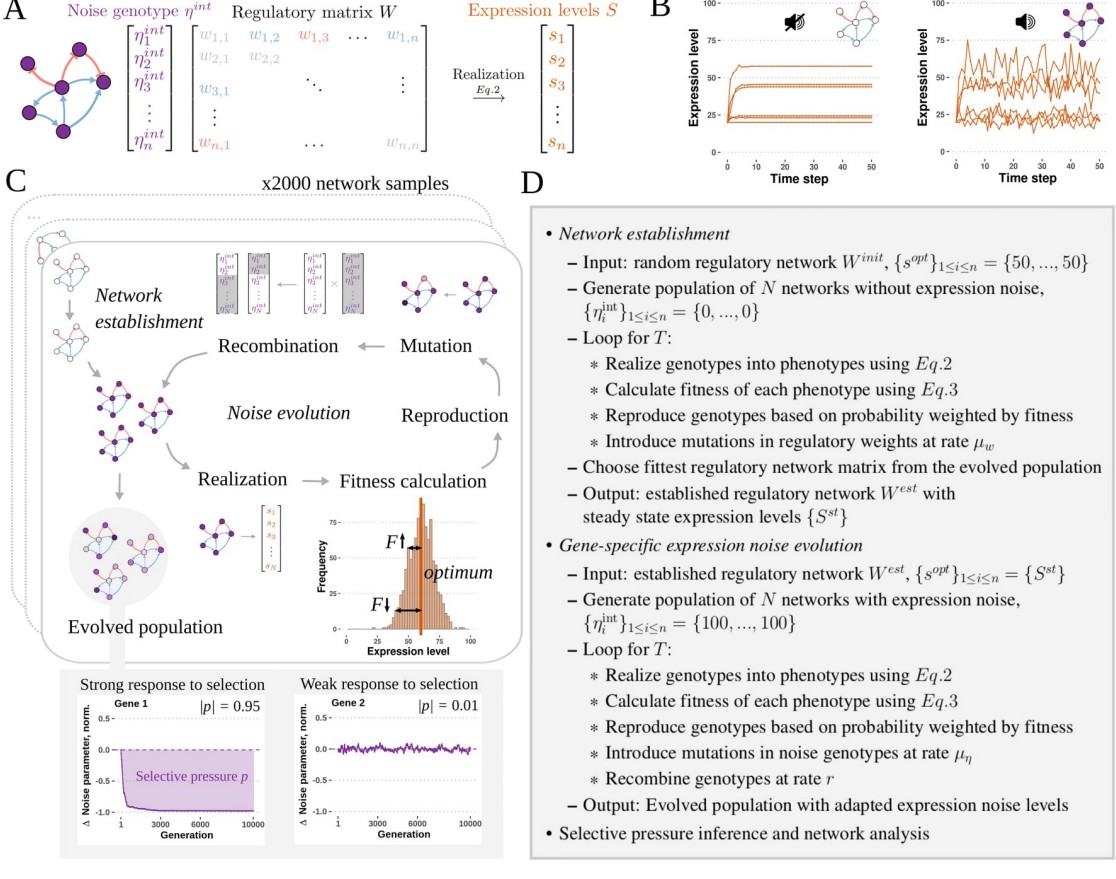

**Fig 1. The evolution of gene-specific expression noise was simulated using populations of model gene regulatory networks with mutable levels of gene-specific expression noise under selective and non-selective conditions. A**—Gene regulatory network model. The genotype consists of the intrinsic noise vector $\eta^{int}$ and regulatory matrix $W$. The intrinsic noise vector defines the gene-specific expression variance of each gene in the network. The regulatory matrix defines the regulatory interactions in the network. The genotype is realized into the phenotype using the dynamical equation described in the main text. The phenotype is given by the state vector $S$, which represents the expression level of each gene in the network. **B**—Deterministic (left) and stochastic (right) realizations of the model. **C**—Steps of the evolutionary simulation process. Each established network configuration was used as a founding network for the network populations used in the noise evolution simulation. In every generation, genotypes are realized and phenotypes (expression levels) are sampled from the last time step. Fitness is calculated from the expression levels. If the populations are evolved under selection, fitness is calculated as the distance of the expression level of each gene from the optimal expression level. Genotypes are reproduced based on their relative fitness and mutations in the intrinsic noise vectors are introduced. Noise genotype vectors are recombined by randomly choosing individuals for recombination and shuffling their noise vectors. The process is repeated for 10,000 generations. **D**—Algorithm overview.

in this study). In every time step $t$ ($1 \leq t \leq T_r$, with $T_r = 50$ in this study), the expression level of each gene is recomputed. The cumulative effect of all transcription factors in the expression level of each gene is for simplicity considered to be additive, *i.e.* we assume there is no cooperative or competitive binding of transcription factors to transcription factor binding sites. This assumption removes the small degree of non-linearity in the response of the regulated gene to transcription factor concentrations, which is present in real transcription factor regulation dynamics. The activation rate $a_i(t)$ is defined as the sum of all effects the regulators of gene $i$ have on its expression level at time step $t$:

$$a_i(t) = \sum_{j=1}^{n} w_{ij} \cdot s_j(t), \tag{1}$$

in which case the dynamic equation for the expression level of each gene in the following time step is:

$$s_i(t + 1) \sim \mathcal{N}(s_i^{basal} + a_i(t), \eta_i^{int}). \tag{2}$$

In every time step the expression level of a gene is drawn from a random distribution. We implemented a simple Gaussian noise, where the mean of the normal distribution equals the sum of basal expression level ($s_i^{basal}$) and activation rate ($a_i(t)$), and the variance equals the gene noise genotype ($\eta_i^{int}$). If the expression level value drawn from the normal distribution is below the minimal ($s_{min} = 0$) or above the maximal expression level ($s_{max} = 100$), it is set to the minimal or maximal expression level, respectively. We note that the shape and variance of the distribution is constant in realization time in our model, but that the expression levels of each individual is the product of the trajectory of the expression levels during the realization process, during which expression levels can exhibit phenotypic switching between stable states. Consequently, there can be a non-normal expression level distribution of a certain gene in the clonal population, even though the expression levels in each time step are drawn from a normal distribution.

The expression levels of all genes are synchronously updated in each time step. The steady state expression levels are invariant to whether the expression levels of each gene are updated synchronously or asynchronously (S1 Text). Similarly, mean expression level, expression variance, CV, noise and Fano factor are invariant to the updating mode (S1 Text). The model may be realized as stochastic or deterministic, depending on the noise parameter values (Fig 1B). The deterministic realization has been used to benchmark the model and to set up the mean expression levels for the starting populations, and the stochastic realization has been used in the main bulk of the simulations, in which intrinsic noise is evolved.

## Forward-in-time simulation of expression noise evolution

To investigate how gene-specific expression noise of constituent genes responds to stabilizing selection at the network level, we used the newly introduced model to perform forward-in-time evolutionary simulations in which we allow the gene-specific noise levels to mutate. An *in silico* evolutionary process consisting of rounds of mutation, selection, recombination and replication events of a population of $N$ ($N = 1, 000$ in this study) individuals was performed for $T$ ($T = 10, 000$) generations (Fig 1C).

We first generated network topologies that would serve as the founding network for the populations in our simulations. We generated 2,000 random (Erdős–Rényi model) network topologies of 40 nodes with regulatory strength values drawn from a uniform distribution $\mathcal{U}(-3, 3)$. The network density was $d = 0.05$. Only connected network graphs were used, meaning there is only one component and there are no disconnected subgraphs.

Autoregulation is not present, because it affects gene-specific noise levels and would be a confounding factor in the analysis. In order to assess the effect of the topology structure on the evolution of expression noise, we also generated an additional 1,000 scale-free (Barabási–Albert model) and 1,000 small-world (Watts–Strogatz model) network topologies with the same size and density. Both random and small-world networks are characterized by a Poisson degree distribution and short mean shortest path length, but random networks have a low clustering coefficient, while small-world networks have a high clustering coefficient. Scale-free networks are characterized by a degree distribution that follows a power law. Real-world networks exhibit degree distributions similar to power-law distributions, high clustering and short path lengths. As such, real-world networks have features of both scale-free and small-world networks [21].

In the simulation of expression noise evolution the regulatory interactions were immutable and the values of the noise genotype vectors were allowed to mutate. Stabilizing selection, the selection scenario in which individuals with extreme phenotypic values have a lower fitness, was imposed on all constituent genes by setting the value of optimal expression level as the mean equilibrium expression level of each gene. The fitness $F(s)$ of a phenotype $s$ was calculated as in Laarits et al. [22], where fitness is defined as the distance from the optimal expression state vector $\{s_i^{opt}\}_{1 \leq i \leq n}$, weighted by the fitness contribution given by $\{\rho_i\}_{1 \leq i \leq n}$:

$$F(s) = e^{-\sum_{i=1}^{n} |s_i^{opt} - s_i|/n\rho_i} \tag{3}$$

The fitness contribution parameters $\{\rho_i\}_{1 \leq i \leq n}$ define the contribution of each gene to the fitness of the phenotype, *i.e.* it is a scaling factor of the decrease of fitness as a function of the distance of the expression level from the optimal expression level for each gene. In this study, the strength of the imposed selective pressure is set to be identical for all constituent genes ($\forall i \, \rho_i = 1$). The assumption of all genes having identical fitness contribution is biologically unrealistic, so we have also performed simulations in which we impose unequal fitness contributions among genes in the same network. We found consistent conclusions (S5 Text), and, for simplicity, we report the results with equal fitness contributions here. Since the fitness contribution of all genes is identical, any differences in the evolutionary outcome we observe after removing the effect of drift will be due to gene differences in their network interactions. Individuals were reproduced into the next generation with a probability equal to their relative phenotype fitness. The fitness of all phenotypes in populations evolved under non-selective conditions was set to an equal constant value, regardless of gene expression levels. Mutations were introduced at a rate $\mu_\eta$ ($\mu_\eta = 0.01$) per gene per replication event. The values for noise genotype mutations were drawn from a normal distribution $\mathcal{N}(100, 40)$. There is no experimental evidence for the shape of the distribution of the expression noise and regulatory strength mutations. We chose a normal distribution because: 1) it defines equally frequent beneficial and deleterious mutations and 2) most mutations would have a small effect, which reflects the characteristic of many studied distributions of fitness effects in model organisms. Recombination was implemented by choosing a random offspring individual at a rate $r$ ($r = 0.05$) and introducing a random break point in the linear genome. The genotype values in the genome segment defined by the break point were then exchanged with another randomly chosen individual from the offspring population. A constant population size $N$ ($N = 1,000$) was maintained. To account for the effect of genetic drift, the noise evolution simulations of each founding network population were replicated 10 times under selection and 10 times under neutrality.

We found that the expression level of most genes in networks with random configurations converge to either $s_{min}$ or $s_{max}$ under a deterministic realization. The measurement of variance of genes that are either not expressed at all or expressed at the maximal level would be impaired since their expression range is constrained by the lower and upper expression level boundary. Since the study of expression variance is our main focus, we added a network establishment step before the noise evolution simulations, in which we subject the network regulatory matrix to mutation and selection for intermediate expression levels. During the network establishment step networks are realized deterministically, *i.e.* the intrinsic noise genotype of all genes is 0. Networks with intermediate steady state expression levels were established through the evolutionary process by imposing a target expression level $\{s_i^{opt}\}_{1 \leq i \leq n}$ ($\{s_i^{opt}\}_{1 \leq i \leq n} = \{\frac{s_{max}}{2}, \ldots, \frac{s_{max}}{2}\}$) for all genes and allowing the strength of regulatory interactions to mutate. Mutations were introduced at a rate $\mu_w$ ($\mu_w = 0.1$) in non-zero entries in the regulatory matrix, preserving the network topology structure (Erdős–Rényi, Barabási–Albert, or Watts–Strogatz model). The values for regulatory strength mutations were drawn from a normal distribution $\mathcal{N}(0, 2)$. Recombination was not implemented at this stage. Fitness of each individual was computed as the distance of the phenotype to the optimal expression state vector using Eq 1. Individuals were reproduced with a probability equal to the relative fitness and the population size kept constant. Network regulatory configurations in which the expression level of all genes would not converge to a fixed point and would oscillate were discarded, as in previous studies [22]. Oscillating gene expression level patterns create population-level heterogeneity generated by the system oscillations and not by stochastic gene expression. Since we are studying the evolution of gene-specific expression noise, expression noise generated by oscillations would be a confounding factor in our analysis. We note, however, that oscillatory networks can be frequent in simulations [23] and biological systems [24], and the role of expression noise in their behavior is an interesting perspective for follow-up studies. Expression level dynamics were termed oscillating if the sum of the differences between expression level in the last time step and previous $\tau$ time steps ($\tau = 10$) was higher than $\epsilon$ ($\epsilon = 10^{-6}$). A stable, *i.e.* non-oscillating, expression level dynamics satisfied the following criterion [22]:

$$\Phi(S(t)) = \frac{1}{\tau} \sum_{\theta=t-\tau}^{t} D(S(\theta), S(t)) < \epsilon \tag{4}$$

where $D$ is the distance between two vectors $D(S^1, S^2) = \sum_{i=1}^{n} |S_i^1 - S_i^2|/n$.

The network establishment process consisting of rounds of mutation, selection and reproduction of a population of $N$ ($N = 1,000$) individuals was performed for $T$ ($T = 10,000$) generations, for each network topology. At the end of the network establishment process, 68% (54333/80000) of genes had intermediate expression levels (S1 Text). The reason why a minority of the genes do not reach close to optimum expression levels could be potential network configuration constraints or a non-extensive optimization/fitting algorithm. Genes that had an expression level of 0 or $s_{max}$ were filtered out from the dataset used in the final analysis. The network regulatory configuration with the highest fitness was chosen from the evolved population and this network configuration was used to generate the starting population for the noise evolution simulations.

The gene network model and evolutionary simulations were implemented in C++ and the source code is available at https://gitlab.gwdg.de/molsysevol/supplementarydata_ expressionnoise/cpp.

## Analysis of simulation results: Expression noise and network centrality measures

The evolutionary outcomes (*i.e.* the change of phenotypes and genotypes) were measured as change of expression noise and selective pressure for each network, respectively. Expression noise in the first and last generation in each evolved population was measured as the variance of the population expression level states for each gene. The change of expression noise (phenotypic evolution) between the first and last generation was measured as the relative change of expression noise, calculated as the difference of expression variance between the first and last generation divided by their sum $(\sigma^2_{gen1} - \sigma^2_{gen10k})/(\sigma^2_{gen1} + \sigma^2_{gen10k})$.

The selective pressure (genotypic evolution) acting on each gene was measured as the average change of noise genotype in every second generation relative to the starting level (Fig 1C). To compare the effect of node centrality on the selective pressure acting on constituent genes, we computed node-level network centrality measures for each node in the networks. We focused our analysis on two local network centrality measures, node instrength and outstrength, but over 30 network centrality measures were analyzed (S2 Text). Instrength of node $i$ is measure of the strength and number of in-going links, *i.e.* how strongly a gene is being regulated:

$$\text{Instrength}(i) = \sum_{j}^{n} |w_{ij}|. \tag{5}$$

Conversely, the outstrength of node $j$ is a measure of the strength and number of outgoing links, *i.e.* how strongly a gene regulates other genes downstream:

$$\text{Outstrength}(j) = \sum_{i}^{n} |w_{ij}|. \tag{6}$$

Further, we computed global graph-level metrics, such as mean graph distance and performed a principal component analysis to reduce the dimensionality (S2 Text). The results were analysed in R 3.6.3 [25]. Network analyses were performed using the `igraph 1.2.4.2` [26] and `statnet 2019.6` [27] packages. Principal component analysis was performed using the `ade4 1.7.15` [28] package.

## Analysis of simulation results: Linear modeling

We fitted linear mixed-effects models using network centrality measures as fixed effect variables and the network topology sample as a random effect variable, allowing for control of intra-network correlation in the response variable. We tested different transformations of the response and explanatory variables in order to improve linearity, and variance structures to account for heteroskedasticity of the residuals. A model where the residual variance was an exponential function of the node absolute instrength was shown to provide the best fit according to the minimal Akaike's Information criterion and was used for all subsequent models (S3 Text). Two types of models were fitted: a logistic regression where the response variable was set to whether a gene answered to selection or not, and standard regressions that used expression variance, relative change of expression variance or selective pressure as response variables. Linear mixed-effect modelling was performed using the `nlme 3.1.144` [29] and `lme4 1.1.27.1` [30] packages. Marginal and conditional $R^2$ values were computed using the `MuMIn 1.43.17` [31] package. Network centrality measures used as explanatory variables in our linear models were correlated (Pearson's $r = -0.17$, p-value $< 2.2 \times 10^{-16}$, S2 Text), so we computed the variance inflation factor (VIF) using the `car 3.0.11` [32]

package. The VIF of all linear models was less than 3; therefore, colinearity was considered to have negligible impact on the inferred statistical significance [33]. To improve homoskedasticity of the residuals in the linear models, we also performed each model fit on two filtered datasets: one in which genes with zero values of instrength or outstrength were removed, and one in which only genes with zero values of instrength or outstrength were kept. The same pattern of effects and significance is observed in the filtered as in the main dataset, so we included the results of the complete dataset in the main text and reported the results of the reduced dataset in the supplementary information (S6 Text).

Finally, since in some cases variable transformation, heterogeneous variance modeling and data filtering did not ensure normality and independence of the residuals, we assessed the amount of resulting bias in the estimation of p-values using a randomization test, in which we fitted a selected model on 10,000 permuted datasets. We chose the model of relative noise change (S3 Text), as the corresponding residuals were significantly departing normality (Shapiro-Wilk test, p-value $< 2.2 \times 10^{-16}$) and independence (Box-Ljung test, p-value $= 8.9 \times 10^{-7}$). For each permutation, we shuffled the values of the response variable (relative change of variance) within each network topology, which removes the effect of network metrics on the change of noise, but preserves the distributions of each metric per network, as well as putative colinearity between explanatory variables. Using $\alpha = 0.05$ as a significance cutoff value, we found a false discovery rate (FDR) of 6.0% for the effect of instrength and and 6.7% for the effect of outstrength. While these values are above the expected 5%, the FDR inflation was found to be relatively low and we concluded that the non-normality of residuals did not affect our conclusions.

### Analysis of simulation results: Information-based metrics

Generalized linear mixed-effects models make several assumptions that might be violated by the data in some cases. Namely, they assume a normal distribution and homoskedasticity of Pearson's residuals, and a normal distribution of random effects. To further validate our conclusions, we computed the mutual information (MI) between variables, which does not have any prior assumptions. We calculated mutual information between the expression noise and centrality metrics using the `infotheo 1.2.0` [34] package. Monte Carlo permutation tests with 10,000 permutations were used to compute p-values for the significance of the mutual information between each pair of tested variables.

## Results

We investigate how selection at the gene network level may lead to the evolution of differential gene-specific expression noise, as observed in biological systems. To do so, we introduce a new gene regulatory model with stochastic gene expression, which extends Wagner's model [20] by adding node-specific intrinsic noise parameters (Fig 1A and 1B). In this framework, the phenotype is represented by the expression level of each gene, and is the realization of a random distribution determined by the genotype. The fitness of an individual is further determined by its distance to an optimal phenotype, therefore, stabilizing selection is implemented as acting on the expression level. We used this model to simulate the evolution of populations of gene regulatory networks with mutable levels of gene-specific expression noise under selective and non-selective conditions (Fig 1C and 1D), and assessed how node properties affect the evolution of intrinsic noise.

### Expression noise propagates along the regulatory network

We first investigated how noise propagated in the model gene regulatory networks. It was shown that noise is additive in biological networks and, therefore, propagates from regulators

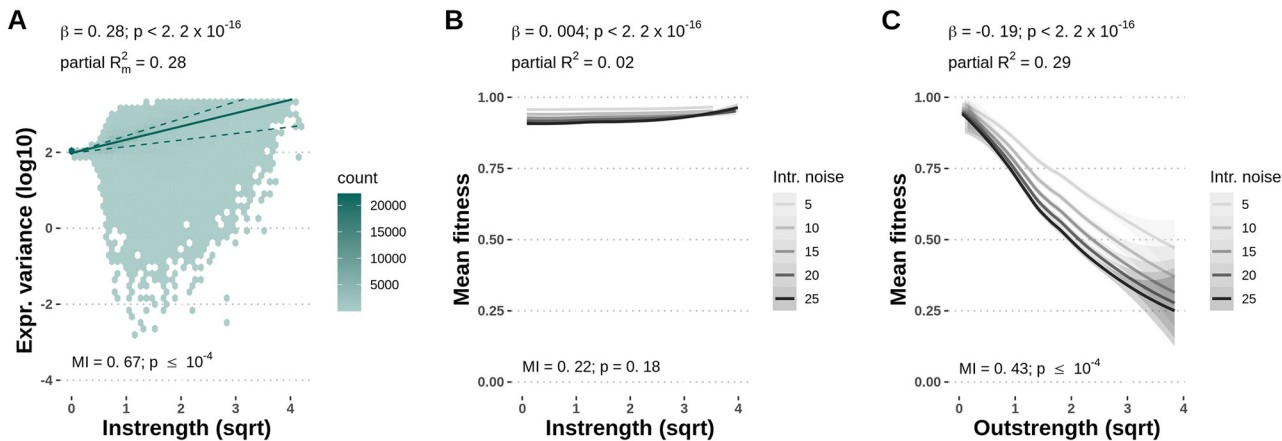

**Fig 2. Noise propagation is captured by the gene regulatory network model. A**—Gene-specific expression variance increases with the absolute instrength of the node, indicating noise propagation is reflected in the gene regulatory network model. The lines indicate the 25% (lower dashed line), 50% (solid line), and 75% (upper dashed line) fitted quantiles. **B, C**—Gene-specific expression variance decreases fitness in gene networks under stabilizing selection on gene expression level. Increasing the level of gene-specific expression noise reduces the mean fitness of the clonal population. The mean fitness of the population is significantly, but marginally, increased by noise in genes with higher node instrength (B), and significantly decreased by noise in genes with higher node outstrength (C). Lines represent the smoothed conditional means and grey bands represent the 95% confidence interval bands. Coefficients, p-values and partial marginal $R^2$ measures are estimated using linear mixed-effects models with expression variance or mean fitness as the response variable, instrength and outstrength as fixed effect explanatory variables, and the network topology sample as the random effect explanatory variable. Mutual information (MI) p-values were computed with a permutation test with 10,000 permutations.

to regulated genes [17, 18]. To assess whether our model successfully captured this property, we generated a dataset of 2,000 realized random network topologies, and tested whether gene expression variance increased with the number of ingoing regulatory links. As expected, we found that the absolute instrength of a gene had a significant positive effect on gene expression variance (linear mixed-effects model with coefficient $\beta = 0.28$, p-value $< 2.2 \times 10^{-16}$) (Fig 2A), indicating that noise propagation was captured in our model. Furthermore, the mutual information between gene expression variance and absolute instrength was significant (MI = 0.67, p-value $\leq 10^{-4}$, permutation test). High node instrength increases expression noise, in line with the experimental evidence that the noisiness of promoters increases with the number of regulatory inputs [35].

We then looked at fitness costs associated with high expression noise in regulators and regulated genes. In a dataset of 1,000 random network topologies, we assessed the mean fitness of the clonal populations of 1,000 individuals under stabilizing selection on the expression level. Each gene was imposed 5 different levels of intrinsic noise, while the intrinsic noise of the rest of the network was kept at 0. We found that increasing the level of expression noise of a single gene decreased the mean fitness of the network (linear mixed-effects model with coefficient $\beta = -0.002$, p-value $< 2.2 \times 10^{-16}$), as expected. However, the strength of this effect depended on the gene centrality. The reduction of fitness due to gene-specific expression noise was significantly, but marginally, affected by instrength (linear model with coefficient $\beta = 0.004$, p-value $< 2.2 \times 10^{-16}$, Fig 2B). The mutual information between mean fitness of the population and absolute instrength was not significant (MI = 0.22, p-value = 0.18, permutation test). However, the mean fitness significantly decreased with node outstrength (linear model with coefficient $\beta = -0.19$, p-value $< 2.2 \times 10^{-16}$, Fig 2C). The mutual information between mean fitness of the population and absolute outstrength was significant (MI = 0.43, p-value $\leq 10^{-4}$, permutation test). Higher fitness cost of expression noise in gene with high outstrength suggests there is a differential selective pressure acting

on genes based on their centrality in the gene regulatory network, which we explore in the next section using an *in silico* evolutionary experiment.

## Gene expression noise is reduced under a stabilizing selection regime

To investigate how gene-specific expression noise responds to stabilizing selection at the network-level, we simulated the evolution of 2,000 random network topologies with and without selection on the gene expression level. We observed that gene expression variance decreased throughout evolution under selective conditions (Fig 3A), and the distribution of intrinsic noise parameters in the population shifted towards lower noise genotype values (Fig 3B), indicating that low-noise alleles conferred a fitness increase to the network. Conversely, gene expression variance remained constant throughout evolution under neutral conditions, and the distribution of noise genotypes reflected only the distribution of random mutations. Replicating the simulations for each network topology sample yielded similar reduction of gene expression variance (Fig 3C) and median noise parameter in the population (Fig 3D). As the initial networks were at their optimal expression level, the mean expression level did not change during evolution and was highly correlated between the first and last generations (Pearson's r = 0.99, p-value $< 2.2 \times 10^{-16}$, S1 Text), confirming that selection acted only on the gene expression variance. Population size had a positive effect on the selective pressure acting on genes, as expected, selection being more efficient in large populations (S1 Text). A population size of 1,000 individuals was chosen for the main simulations as the optimal population size in the trade-off between selecting mutations with small effects and reducing computational speed.

Next, we investigated how individual nodes within a network respond to selection, based on their centrality properties.

## Evolutionary change in phenotypes: Regulators reduce their expression noise to a higher degree

We first analysed the phenotype change, *i.e.* the relative change in gene-specific expression variance after evolution. The variance of gene expression depends both on the intrinsic noise

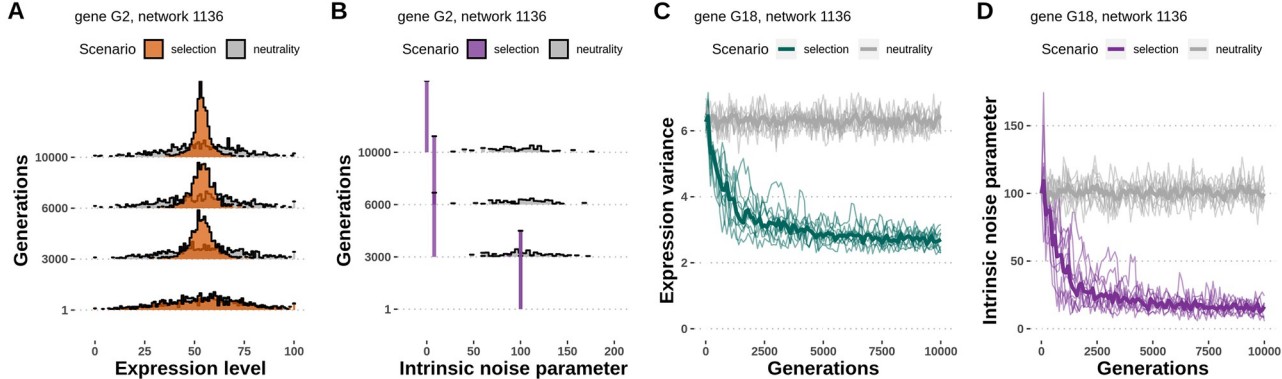

**Fig 3. Gene-specific expression noise evolves in a model with selection. A**—The distribution of expression levels of an example gene throughout evolution in populations evolved under stabilizing selection on gene expression level and under neutrality. The variance of gene expression level is reduced under selection, but not under neutrality. **B**—The distribution of intrinsic noise parameters of an example gene throughout evolution in populations evolved under selection and under neutrality. The median intrinsic noise parameter skews to lower values under stabilizing selection, but not under neutrality. **C, D**—Replicates of the simulations with the same input network and parameters. Replicates have different dynamics, but reach similar outcomes in terms of expression variance (C) and median intrinsic noise parameter (D) in the evolved populations. The evolution of each network topology sample was replicated 10 times under selection and 10 times under neutrality.

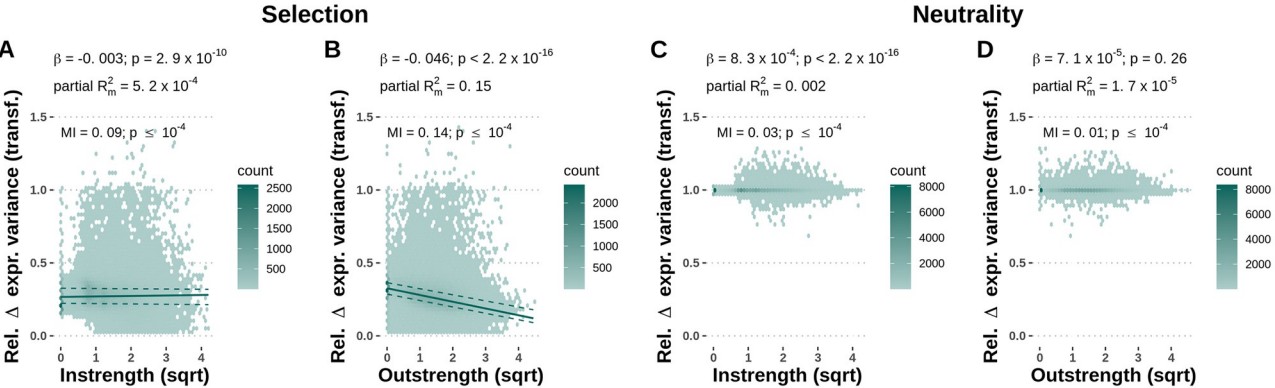

**Fig 4. Node-level network centrality measures affect the relative change of gene-specific expression variance under network-level selection.** For each gene, the relative change of expression variance before and after evolution (Rel. Δ expr. variance) was averaged over all replicates. **A, B**—Absolute instrength (A) and absolute outstrength (B) have a significant negative effect on the relative change in gene expression variance in populations evolved under selection. A lower value of relative change of expression variance indicates a bigger reduction in expression variance between the first and last generation and a stronger response to selection. The lines indicate the 25% (lower dashed line), 50% (solid line), and 75% (upper dashed line) fitted quantiles. **C, D**—Absolute instrength (C) and absolute outstrength (D) have a significant, but negligible, negative effect on the relative change in gene expression variance in the populations evolved under neutrality. The dataset consists of 74,443 genes from 2,000 populations with unique 40-gene random network topology samples, which were independently evolved 10 times under selection and 10 times under neutrality. Coefficients, p-values and partial marginal $R^2$ measures were estimated using linear mixed-effects models with relative change of gene-specific variance as the response variable, instrength and outstrength as fixed effect explanatory variables, and the network topology sample as the random effect explanatory variable. Mutual information (MI) p-values were computed using 10,000 permutations.

of the genes (that is, its genotype in our model) and the number and noise of the genes it is connected with.

We fitted linear models to assess the impact of the absolute instrength and outstrength measures on the relative change in expression variance for each node in each network. Under selection, both absolute instrength and outstrength had a significant negative effect (linear mixed-effects model with coefficients $\beta_{\text{instrength}}$ = -0.003, p-value = 2.9 × $10^{-10}$, Fig 4A; $\beta_{\text{outstrength}}$ = -0.046, p-value < 2.2 × $10^{-16}$, Fig 4B), meaning that genes with more and stronger connections reduced their expression variance to a larger extent than less connected genes. The effect was notably stronger for outstrength (marginal $R^2$ = 0.15) than for instrength (marginal $R^2$ = 5.2 × $10^{-4}$). Similarly, the mutual information was significant between the relative change in gene expression variance under selection and absolute instrength (MI = 0.09, p-value $\leq 10^{-4}$, permutation test) and absolute outstrength (MI = 0.14, p-value $\leq 10^{-4}$, permutation test). Genes with high outstrength are strong regulators and their reduction of expression variance to a larger extent indicates that high expression noise is more detrimental in regulators than in regulated genes. Under neutrality, absolute instrength had a significantly positive effect (linear mixed-effects model with coefficient $\beta$ = 8.3 × $10^{-4}$, p-value < 2.2 × $10^{-16}$, Fig 4C) and absolute outstrength did not have a significant effect on the relative change in gene expression variance (linear mixed-effects model with coefficient $\beta$ = 7.1 × $10^{-5}$, p-value = 0.26, Fig 4D). The mutual information was significant between the relative change in gene expression variance under neutrality and absolute instrength (MI = 0.03, p-value $\leq 10^{-4}$, permutation test) and absolute outstrength (MI = 0.01, p-value $\leq 10^{-4}$, permutation test). These effects are much smaller and of opposite direction than the ones measured in selective conditions, indicating that genetic drift did not cause the effect of centrality measures on expression variance observed in selected populations.

### Evolutionary change in genotypes: Regulators are more likely to respond—And display a stronger response—To selection

To investigate differential selective pressure acting on gene-specific expression noise, we analysed the change of intrinsic noise parameters in populations of gene regulatory networks evolved with or without stabilizing selection on the expression level. We measured the selective pressure acting on individual genes as the average reduction in the intrinsic noise parameter relative to the beginning of the evolutionary simulation (see Methods). The selective pressure on genes was found to be close to 0 in neutrally evolving populations, as expected (Fig 5B). In the presence of selection, however, the distribution of selective pressures was found to be bimodal (Fig 5A). Therefore, we binned genes in two categories according to whether they responded to selection (selective pressure > 0.5) or not (selective pressure ≤ 0.5). We then separately analysed the probability to respond to selection and the strength of the response.

Absolute instrength had a significant and strongly negative effect (logistic regression with coefficient $\beta$ = -1.87, p-value < $2.2 \times 10^{-16}$, Fig 5C) on the probability of a gene to respond to selection, that is, genes with more and stronger incoming links are less likely to respond to selection. Absolute outstrength also had a significant effect on the probability of a gene to respond to selection (logistic regression with coefficient $\beta$ = -0.08, p-value = $6.7 \times 10^{-7}$, Fig 5D). However, this effect was small and was lost when the interaction terms between instrength and outstrength were included in the model (SI).

For a qualitative analysis of the effect of network centrality on the selective pressure acting on individual genes, we fitted linear-mixed effects models on the set of genes that responded to selection, with selective pressure as the response variable. In the genes that responded to selection from the selected populations, absolute instrength had a significant negative effect (linear mixed-effects model with coefficient $\beta$ = -0.04, p-value < $2.2 \times 10^{-16}$, Fig 5E). Conversely, absolute outstrength had a significant positive effect (linear mixed-effects model with coefficient $\beta$ = 0.03, p-value < $2.2 \times 10^{-16}$, Fig 5F) on the selective pressure. In the selected populations, the mutual information was significant between the selective pressure and absolute instrength (MI = 0.19, p-value ≤ $10^{-4}$, permutation test) and absolute outstrength (MI = 0.31, p-value ≤ $10^{-4}$, permutation test). In the neutral populations, neither absolute instrength nor absolute outstrength had a significant effect (linear mixed-effects model with coefficient $\beta_{instrength}$ = $2.4 \times 10^{-8}$, p-value = 0.99, Fig 5G; $\beta_{outstrength}$ = $-1.2 \times 10^{-5}$, p-value = 0.49, Fig 5H) on the selective pressure. Similarly, the mutual information was not significant between the selective pressure and absolute instrength (MI = 0.005, p-value = 0.34, permutation test), nor absolute outstrength (MI = 0.005, p-value = 0.45, permutation test).

The increased selective pressure in genes with high outstrength (strong regulators) can be explained by noise propagation to downstream elements. Namely, expression noise in regulators propagates to the genes they regulate, increasing the overall expression noise in the gene regulatory network. If gene expression levels in the network are under stabilizing selection, expression noise is deleterious. Therefore, regulator genes experience a comparatively higher selective pressure to reduce expression noise than regulated genes. In a genome-wide expression noise screen in *Drosophila melanogaster*, transcription factors were found to have lower expression variation [36]. Suppression of expression noise can be attained through negative autoregulation [37–39], whereby a regulator acts as its own repressor. Incidentally, 40% of transcription factors in *E. coli* [40] and many eukaryotic transcription factors [41] have negative autoregulation, indicating a wide-spread control of expression noise in natural regulatory networks.

In contrast to regulator genes, we found that regulated genes, *i.e.* genes with high node instrength, are less likely to respond to selection and the selective pressure decreases with node instrength. Since the expression noise of genes is a sum of their intrinsic noise and noise

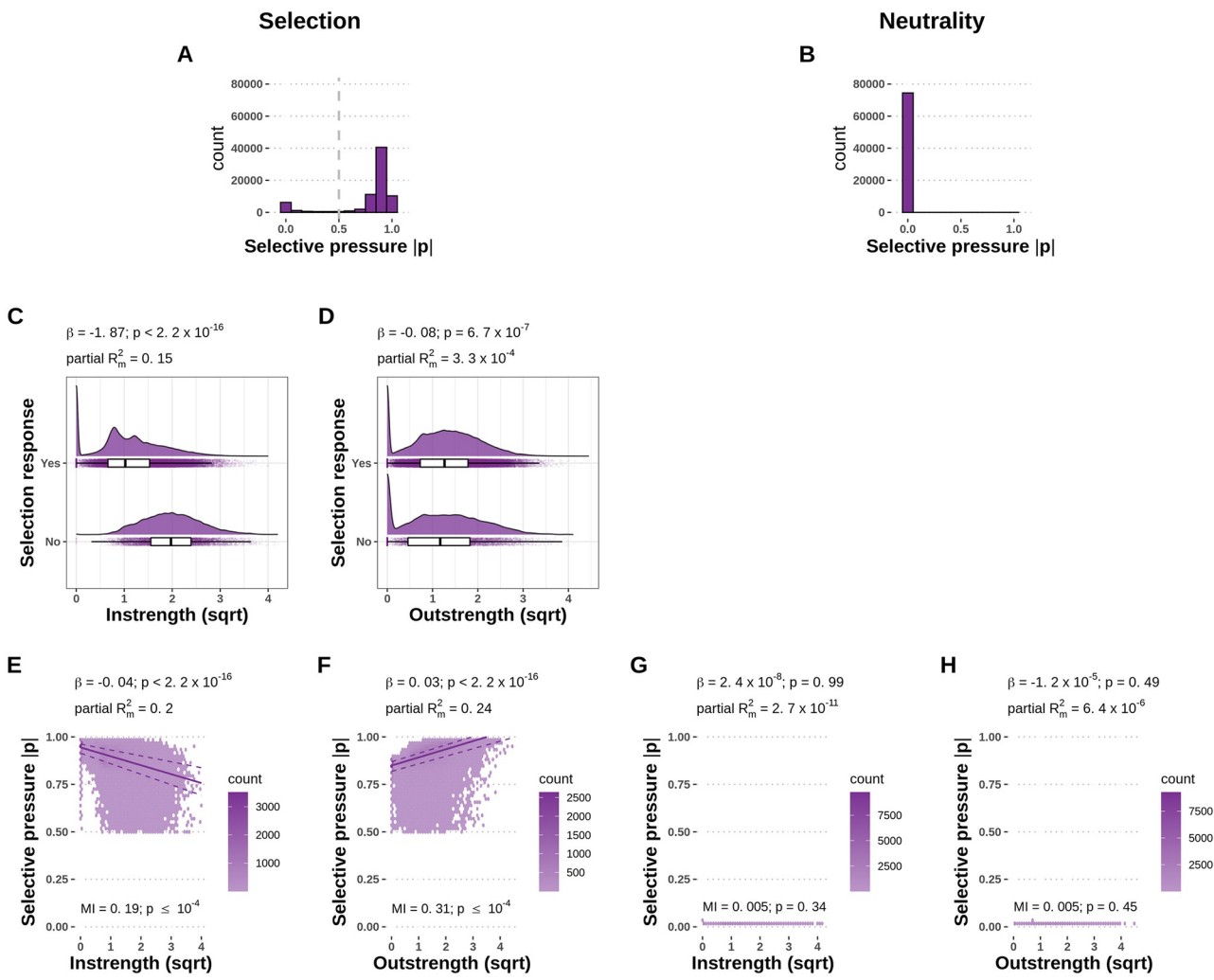

**Fig 5. Differential selective pressure is acting on genes based on their centrality. A, B**—Distributions of the measured selective pressure in selected (A) and neutral (B) populations. Genes with a selective pressure above 0.5 were categorized as responsive to selection. **C, D**—High instrength genes are less likely to respond to selection. Absolute instrength (C) has a strong significant negative effect on the probability of selection response. Absolute outstrength (D) has a weak significant negative effect on the probability of selection response. **E, F**—In the subset of genes that responded to selection, high instrength (E) decreases the selective pressure, while high outstrength (F) increases the selective pressure acting on individual genes. The lines indicate the 25% (lower dashed line), 50% (solid line), and 75% (upper dashed line) fitted quantiles. **G, H**—Absolute instrength (G) and outstrength (H) have no significant effect on the selective pressure in the non-selected populations. The dataset consists of 74,443 genes from 2,000 populations with unique 40-gene random network topology samples, which were independently evolved 10 times under selection and 10 times under neutrality. The selective pressure on each gene is calculated as the average normalized reduction of the intrinsic noise parameter during the evolutionary simulation and summarized as the mean over all replicates in each scenario. Coefficients, p-values and partial marginal $R^2$ measures are estimated using logistic regression and linear mixed-effects models with selection responsiveness or selective pressure as the response variable, instrength and outstrength as fixed effect explanatory variables, and the network topology sample as the random effect explanatory variable. Mutual information (MI) p-values were using 10,000 permutations.

propagated from upstream elements, the contribution of intrinsic noise to the total noise of the gene will be comparatively smaller in strongly regulated genes. The network can thus respond to selection either by reducing the intrinsic noise of the focal gene, or by reducing the intrinsic noise of any of the upstream elements, which would reduce propagated noise. As a result, there is a relaxation of selective pressure in regulated genes, which is distributed on upstream genes. On the other hand, the same mechanism increases the selective pressure on upstream genes, *i.e.* regulators.

To check the robustness of our results, we performed the node-level network centrality analysis on two additional datasets with different topology structures: scale-free (Barabási–Albert) and small-world (Watts–Strogatz) topology models. We find consistent effects (direction and significance) of local network centrality metrics on the selective pressure acting on gene-specific noise across topology models, showing that our findings are robust to the topology model used (S4 Text). However, the effect size of network centrality metrics differed between the topology models, pointing at an effect of the topology model on noise propagation and the evolution of gene-specific expression noise, which we investigate in the next section.

## Global network properties affect the evolvability of expression noise and selective pressure on constituent genes

Lastly, we analysed how topological structures and graph-level network properties affect the expression noise response of constituent genes to selection on a joint dataset of random (Erdős–Rényi), scale-free (Barabási–Albert) and small-world (Watts–Strogatz) network topologies. Jointly analysing genes from all three topology types with linear models, we observed statistically significant interactions between instrength and outstrength and network topology types on both the probability to respond to selection and the selective pressure acting on gene-specific expression noise (Table 1). We found that genes in scale-free networks have a significantly higher probability of responding to selection than genes in random networks. These results are in agreement with previous studies reporting a higher evolvability of scale-free networks [42, 43]. Conversely, genes in small-world networks have a significantly lower probability of responding to selection than genes in random networks. Furthermore, there are significant effects of interactions between instrength and outstrength with the topology type on the selective pressure on constituent genes.

To investigate which global topological features of the three network models affect expression noise evolution, we performed a principal component analysis (PCA) on 12 graph-level measures. The first two dimensions of the PCA expressed 85.4% of the total dataset inertia (S2 Text), so we used the first two principal components (PCs) as synthetic explanatory variables in linear mixed-effects models. The loading of the first synthetic variable (PC1) is dominated by negative loadings of diameter and mean path distance, and the centralization measures, namely positive loadings of outdegree and closeness centralization and negative loadings of indegree and betweenness centralization. The diameter of a network is defined as the longest shortest path between any two nodes. Centralization is a measure of the extent to which a network is centered around a single node and can be computed from different centrality metrics. The loading of the second synthetic variable (PC2) is dominated by the negative loading of the average degree, average indegree and average outdegree measures (S2 Text). For a more intuitive interpretation, the signs of both PCs have been switched in the statistical analysis. Therefore, PC1 shown in the results is dominated by positive loadings of diameter, mean path distance, indegree centralization and negative loadings of outdegree centralization, and PC2 is dominated by positive loadings of average degree. We refer to PC1 and PC2 as synthetic network diameter and centralization and synthetic average degree, respectively.

The average expression variance per network is significantly negatively affected by synthetic network diameter and centralization (linear model with synthetic network diameter and centralization coefficient $\beta$ = -6.19, p-value < $2.2 \times 10^{-16}$) and significantly positively affected by the synthetic average degree (linear model with synthetic average degree coefficient $\beta$ = 13.26, p-value < $2.2 \times 10^{-16}$). The mutual information was significant between the average expression variance per network and synthetic network diameter and centralization (MI = 0.21, p-value $\leq 10^{-4}$, permutation test) and synthetic average degree (MI = 0.21, p-value $\leq 10^{-4}$,

**Table 1. Network topology type affects the probability of responding to selection and selective pressure on gene-specific expression noise under stabilizing selection on gene expression level.**

| Response | Explanatory variable | Beta | SE | p-value[1] | |
|---|---|---|---|---|---|
| Probability of responding to selection | Instrength | -1.9270 | 0.0284 | $< 2.2 \times 10^{-16}$ | **** |
| | Outstrength | -0.0829 | 0.0226 | $< 2.6 \times 10^{-4}$ | *** |
| | Scale-free (BA) topology[2] | 0.9209 | 0.1075 | $< 2.2 \times 10^{-16}$ | **** |
| | Small-world (WS) topology[3] | -0.2684 | 0.0945 | 0.0045 | ** |
| | Instrength:BA[4] | 0.0120 | 0.0516 | 0.8159 | n.s. |
| | Instrength:WS | 0.0006 | 0.0401 | 0.9873 | n.s. |
| | Outstrength:BA | -0.2947 | 0.0252 | $< 2.2 \times 10^{-16}$ | **** |
| | Outstrength:WS | -0.0728 | 0.0333 | 0.0287 | * |
| Gene-specific selective pressure | Instrength | -0.0377 | 0.0004 | $< 2.2 \times 10^{-16}$ | **** |
| | Outstrength | 0.0347 | 0.0003 | $< 2.2 \times 10^{-16}$ | **** |
| | Scale-free (BA) topology | 0.0019 | 0.0012 | 0.1404 | n.s. |
| | Small-world (WS) topology | 0.0222 | 0.0013 | $< 2.2 \times 10^{-16}$ | **** |
| | Instrength:BA | 0.0143 | 0.0007 | $< 2.2 \times 10^{-16}$ | **** |
| | Instrength:WS | -0.0055 | 0.0006 | $< 2.2 \times 10^{-16}$ | **** |
| | Outstrength:BA | -0.0151 | 0.0003 | $< 2.2 \times 10^{-16}$ | **** |
| | Outstrength:WS | -0.0075 | 0.0005 | $< 2.2 \times 10^{-16}$ | **** |

[1] Coefficients and their significance were computed using linear mixed-effects models (see Methods). The dataset consisted of 3,000 populations with unique 40-gene random, scale-free and small-world network topology samples, which were independently evolved 10 times under selection and 10 times under neutrality. The selective pressure on each gene was calculated as the average normalized reduction of the intrinsic noise parameter during the evolutionary simulation and summarized as the mean over all replicates in each scenario. Genes were termed responsive to selection if their selective pressure was above 0.5. Asterisks indicate statistical significance: n.s.—p-value > 0.05;

*—p-value $\leq$ 0.05;

**—p-value $\leq$ 0.01;

***—p-value $\leq$ 0.001;

****—p-value $\leq$ 0.0001.

[2] Barabási–Albert network model.

[3] Watts–Strogatz network model.

[4] Colons (':') indicate variable interactions.

permutation test). This finding means that global network properties affect the amplification of noise through noise propagation between the genes. Specifically, networks with a lower diameter, mean path distance, indegree centralization, and higher outdegree centralization and average degree, had higher average gene expression variance. In the selected populations, the average selective pressure per network was significantly negatively affected by both synthetic network diameter and centralization and the synthetic average degree (linear model with synthetic network diameter and centralization coefficient $\beta$ = -0.003, p-value = $4.9 \times 10^{-11}$, Fig 6A; synthetic average degree coefficient $\beta$ = -0.009, p-value $< 2.2 \times 10^{-16}$, Fig 6B). The mutual information was significant between the average selective pressure per network and synthetic network diameter and centralization (MI = 0.27, p-value $\leq 10^{-4}$, permutation test) and synthetic average degree (MI = 0.26, p-value $\leq 10^{-4}$, permutation test). This result shows that the average selective pressure acting on gene-specific expression noise in networks decreases with an increase of network diameter, mean path distance, indegree centralization and average degree per network. Conversely, the average selective pressure increases with an increase of outdegree centralization (Fig 6A and 6B). In the populations evolved under neutrality, neither synthetic network diameter and centralization, nor synthetic average degree,

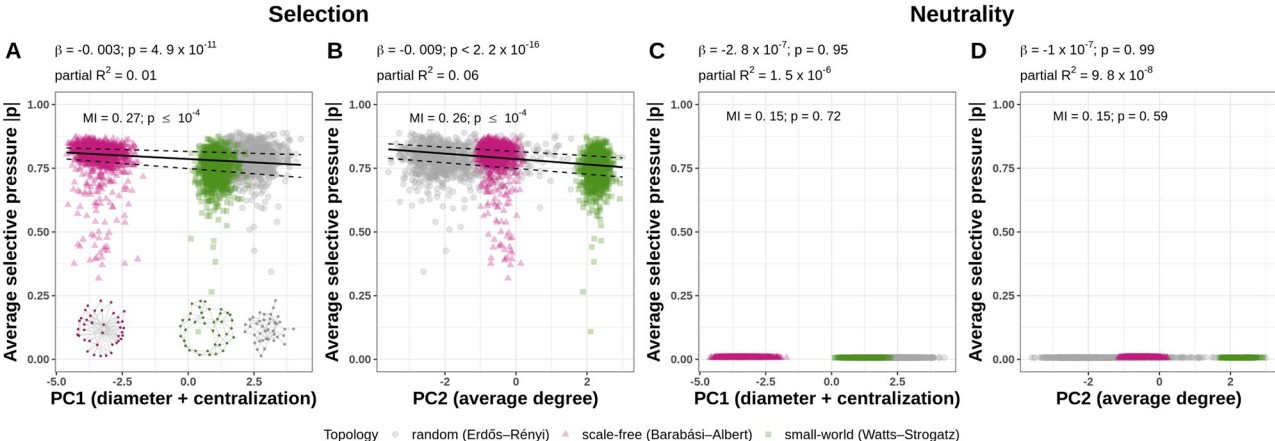

**Fig 6. Global network properties affect the average selective pressure acting on gene expression noise under stabilizing selection on gene expression level. A, B**—Principal component variables consisting of the diameter and network centralization (A) and average degree (B) have a significant negative effect on the average selective pressure per network. The two synthetic variables were constructed by performing a principal component analysis on 12 graph-level network metrics. The lines indicate the 25% (lower dashed line), 50% (solid line), and 75% (upper dashed line) fitted quantiles. The dataset consisted of 3,000 populations with unique 40-gene random, scale-free and small-world network topology samples, which were independently evolved 10 times under selection and 10 times under neutrality. The selective pressure on each gene is calculated as the average normalized reduction of the intrinsic noise parameter during the evolutionary simulation and summarized over all replicates in each scenario. Coefficients and p-values are estimated using a linear model with average selective pressure as the response variable, and PC1 and PC2 as explanatory variables. Mutual information (MI) p-values were computed with permutation test with 10,000 permutations.

had a significant effect on the average selective pressure per network (linear model with synthetic network diameter and centralization coefficient $\beta = -2.8 \times 10^{-7}$, p-value = 0.95; synthetic average degree coefficient $\beta = -1 \times 10^{-7}$, p-value = 0.99, Fig 6C and 6D). Similarly, the mutual information was insignificant between the average selective pressure per network and synthetic network diameter and centralization (MI = 0.15, p-value = 0.72, permutation test) and synthetic average degree (MI = 0.15, p-value = 0.59, permutation test).

## Discussion

In this work, we aimed at understanding how natural selection shaped the distribution of expression noise levels between genes in the genome. We hypothesized that selection for low noise at the network level translates into differential selective pressures at the gene level. To test this hypothesis, we developed a new gene regulatory network evolution model that incorporates stochastic gene expression, where the gene expression mean and variance are both heritable and, therefore, potentially subject to natural selection. We simulated the evolution of gene-specific expression noise in populations of model gene regulatory networks under selective and non-selective conditions. In agreement with our hypothesis, we observed that individual genes respond differently to the global selective pressure and that this response depends on the local and global network properties. In particular, we found that genes of high centrality exhibit a stronger selective pressure to reduce gene-specific expression noise under stabilizing selection on the expression level and that the genetic network structure affects the propagation and evolvability of gene-specific expression noise. In the following, we further discuss the implications of differential selective pressure acting on constituent genes in gene networks.

### Mechanisms of intrinsic noise reduction

In this study we abstracted and summarized the many determinants of intrinsic expression noise into a single parameter, which can be viewed as a modifier locus that can directly change

the intrinsic noise of a given gene. This simplification permitted us to investigate the evolution of expression noise in gene networks with computationally feasible evolutionary simulations. In reality, multiple factors that affect gene expression variance in biological systems have been reported. These include epigenetic factors, such as chromatic dynamics [44] and presence of chromatin remodelling complexes [45]. Other factors affect transcription directly and can, therefore, control expression noise: the promoter shape [36], presence of a TATA box [45], presence and number [4] of TF binding sites, TF binding dynamics [46], presence of TF decoy binding sites [47], and transcription rate. Factors affecting translation have also been shown to play a role in controlling noise: miRNA targetting [48], mRNA lifetime, translation rate, and post-translational modifications such as the protein degradation rate. Compartmentalization of proteins by phase separation has also been shown to reduce noise [49]. Lastly, gene expression costs can also affect the gene expression level distributions, and thereby expression level noise [50]. We have demonstrated the existence of a general selective pressure acting on gene expression noise. Biological organisms may differ in the mechanisms used to respond to this selective pressure, calling for further, data-driven, investigations.

## Global network structure impacts noise propagation and evolution

By simulating thousands of networks with distinct structures, we were further able to assess the impact of global network characteristics on gene-specific selective pressure. Given that there is a trade-off between the fitness advantage of reducing gene-specific expression noise at the gene level and its mechanistic cost (for instance, in terms of mRNA processing [51]), evolving the global network structure may offer an alternative way to reduce network-level noise. Several motifs recurrently found in regulatory networks have an impact on expression noise, such as negative [37–39] and positive autoregulation [41], feed-forward loops [41, 52, 53] and interlinked feed-forward loops [54].

It is important, however, to distinguish two aspects when considering the effect of the network structure on the expression dynamics of constituent genes: the network structure, *i.e.* the topology of the graph, and the strength of each of the regulatory interactions, both of which impact expression noise. The same network topology, but with different regulatory interactions strengths, can give rise to markedly different network behaviours. In the *gap* gene system, for example, it was shown that multiple subcircuits share the same regulatory structure, but yield different expression patterns because of their differences in active components and strength of regulatory interactions [55]. It results that network models of gene expression noise must incorporate both graph topology and interaction strength between all constituent genes. The Wagner model constitutes a simple framework that fulfills these two conditions. However, it has its limitations. Namely, it is not fine-grained enough to capture the complex dynamics of real regulatory networks. Models that incorporate higher molecular detail, such as large systems of differential equations, are necessary to precisely capture in fine detail the expression dynamics of a real biological network, but they come with a cost in terms of high computation time (preventing their use in evolutionary simulations), low tractability and, often, the inability to model noise.

## Implications of selection on expression noise on the evolution of genomes and gene regulatory networks

One mechanism by which networks and genomes evolve is gene duplication. Gene duplications are a major source of new genes and thought to be a primary source of evolutionary novelties. It has been long proposed that new functionality arises from duplicated genes by allowing the other gene copy to acquire new functions (neofunctionalization) or improve existing functions (subfunctionalization) by relaxing the selective pressure acting on a single

gene through an additional redundant copy [56]. However, most of the time the redundant copy is lost before new functionality can arise [57], either by genetic drift alone or because having the extra copy is deleterious. The redundant copy has a chance to evolve a new function or improve an existing one while it is evolving neutrally or reaches fixation in the population, or alternatively, if there is some fitness benefit of the additional copy that increases its frequency in the population. Some benefits of having additional gene copies have been shown, such as increased expression level for genes whose pre-duplication expression level was far from the optimum [58]. Moreover, duplicating a gene reduces its expression noise [59, 60], averaging the stochastic events over the two gene copies. The reduction of expression noise may, therefore, constitute another benefit of a gene duplication, increasing its chance of fixation in the population. As the gene number increases in bacterial genomes, the number of regulatory genes increases 4-fold [61], indicating a gene duplication is more likely to stay if the gene is a regulatory gene. We hypothesize that selection on expression noise, particularly on regulatory genes, could, therefore, be one of the forces driving the maintenance of duplicated genes.

## Applications of the model framework to study complex systems

In this study, we developed a new regulatory and evolutionary model to study expression noise in gene regulatory networks. The model represents key features of evolving gene regulatory networks, namely the non-independence of gene expression levels and fitness determined by the expression level of many or all genes in the network. Our results revealed that differential selective pressure acts on intrinsic expression noise of constituent genes and that network-level topological properties affect noise propagation within the network.

Although our study focused on gene regulatory networks, our conclusions potentially apply to a broader range of systems. In particular, we posit that any system that fulfills two essential properties will exhibit a similar behavior: (i) the amount of product of each system component (here called "expression level") is not independent and (ii) the performance (here termed "fitness") is determined by the product level of one or several of the components of the system. There are many other complex systems that fulfill these criteria, such as biological metabolic networks, ecological food webs, neural networks, economies, transportation and other infrastructure networks, and social networks. We expect that the same constraints act on noise in elements of these systems, and that some of the conclusions from gene regulatory networks could be carefully applied to other complex systems.

## Conclusion

Our results show that selection for low expression noise acting on a system (the gene network) resulted in differential selective pressures on its individual components (the genes). We demonstrated that the position of the gene in the network and the global network structure act as important drivers of the evolution of intrinsic expression noise. Investigating how gene networks evolve to cope with expression noise will reveal mechanisms of how complex biological systems adapt to function with an inevitable molecular noise in their components. A better comprehension of these mechanisms is a prerequisite to understand the evolution of complexity in biological systems, from the first self-replicating RNA systems to modern eukaryotic cells expressing tens of thousands of genes.

## Supporting information

**S1 Text. Gene regulatory network model and evolutionary model.**
(PDF)

**S2 Text. Network centrality metrics.**
(PDF)

**S3 Text. Diagnostics of statistical models.**
(PDF)

**S4 Text. Robustness of results in different topology structures.**
(PDF)

**S5 Text. Robustness of results to unequal fitness contribution of genes.**
(PDF)

**S6 Text. Filtered datasets.**
(PDF)

## Acknowledgments

The authors would like to thank Arnaud Le Rouzic for the beneficial discussions, Andreas Wagner and the Wagner lab for the valuable input during the model development, Arne Traulsen and Tal Dagan for the helpful suggestions throughout the project, Nikhil Sharma for his study on additional aspects of the model, and Andrea Bours, Artemis Efstratiou and Carolina Peralta for the careful reading of the manuscript.

## Author Contributions

**Conceptualization:** Nataša Puzović, Julien Y. Dutheil.

**Formal analysis:** Nataša Puzović, Tanvi Madaan.

**Investigation:** Nataša Puzović, Julien Y. Dutheil.

**Methodology:** Nataša Puzović, Julien Y. Dutheil.

**Software:** Nataša Puzović.

**Supervision:** Julien Y. Dutheil.

**Validation:** Nataša Puzović.

**Visualization:** Nataša Puzović, Tanvi Madaan.

**Writing – original draft:** Nataša Puzović.

**Writing – review & editing:** Nataša Puzović, Julien Y. Dutheil.

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
