## [Decision Letter · Decision Letter 0]

8 Nov 2022

Dear Dr. Puzović,

Thank you very much for submitting your manuscript "Being noisy in a crowd: differential selective pressure on gene expression noise in model gene regulatory networks" for consideration at PLOS Computational Biology.

As with all papers reviewed by the journal, your manuscript was reviewed by members of the editorial board and by several independent reviewers. In light of the reviews (below this email), we would like to invite the resubmission of a significantly-revised version that takes into account the reviewers' comments.

We cannot make any decision about publication until we have seen the revised manuscript and your response to the reviewers' comments. Your revised manuscript is also likely to be sent to reviewers for further evaluation.

Sincerely,

Marija Cvijovic

Guest Editor

PLOS Computational Biology

Kiran Patil

Section Editor

PLOS Computational Biology

Reviewer's Responses to Questions

**Comments to the Authors:**

Reviewer #1: In this manuscript, Puzovic, Madaan, and Dutheil present a stochastic gene-regulatory network to analyze the evolution of gene expression noise in single cells. They build on an existing model, extend it to generate expression noise and benchmark the model through a number of scenarios. The methodology is described rigorously (I miss this in a good fraction of papers), and results should be reproducible and also something to build upon for future work.

Having said that, I do not think that the current version of the manuscript meets the scope and requirements for publication in PLoS Comp Biol. My main concern is that the current results are more of a description/demonstration/benchmarking of the model, and lacks links to biological systems, or new insight overall. I do appreciate, however, the extent of the simulation data generated, yet for me (a bit outside of the "network biology" sub-discipline) this is not enough to convince of the usefulness of such models. I encourage the Authors to think of ways to use their current work for learning new biology (rather than putting a short discussion on putative implications in the Discussion section), and resubmit a head-to-toe improved and updated version.

Reviewer #2: In this manuscript, the authors have studied the evolution of gene-specific expression noise in gene networks. The motivation for this work is to explore how in phenotypes, central genes have shown to have lesser expression noise compared to peripheral genes.

The authors have formulated forward-in-time simulation framework to study the evolution of stochastic gene expression in phenotypes. This is done by imposing stabilization selection and evolving networks through steps of mutation, selection, and recombination. The authors showed that the peripheral genes may have higher expression noise due to increased selective pressure on central genes leading to noise amplification to peripheral genes.

Furthermore, the authors have studied the impact of local and global network features on the expression noise. The authors report higher centrality metrics leads to stronger selective pressure to reduce expression noise.

Overall, this manuscript is interesting, and the methods used are clearly explained. The topic studied in this manuscript is of importance in understanding the evolution of gene networks.

The authors have provided thoughtful supplementary information.

I had a few comments while reading the manuscript.

Is selective pressure defined in eq 3 (“the strength of imposed selective pressure is set to be identical for all constituent genes equals to 1 for all genes” lines 193:194) different from selective pressure defined in line 253 (average change of noise in every second generation)? Lines 193:194 suggests selective pressure is “imposed” to be equivalent to 1 and the same for all.

Line 214: Does under the network establishment step, steady state expression level of genes converge to the optimum level? If they do not, then what may be the reason for not converging to steady state? It may be beneficial to point out this.

Would it be beneficial to define Instrength and Outstrength as the summation of real regulatory strength values \\sum_{j} W_{ij}, as compared to the summation of absolute regulatory strength? If considered summation of real regulatory strength, then it will be beneficial to differentiate between if a gene is overall activated or repressed. I acknowledge this might be out of the scope of this manuscript.

Reviewer #3: The authors extend a previous gene network model and perform corresponding evolutionary simulations to test the hypothesis that the expression noise of highly connected genes in gene regulatory networks is under stronger selective pressure than the expression noise of less connected genes. This work builds on previous studies that have shown that the gene networks modulate expression noise and evolve under selective pressure. The manuscript is well written and the figures are acceptable. Though the main finding is novel, interesting, and timely, I have several major concerns, largely related to the assumptions made in the modelling, that must be addressed before I can accept this manuscript for publication in PLoS Computational Biology.

Major Concerns

Several of the model assumptions are not biologically realistic. For instance, imposing the same selective pressure for all genes ignores the benefits/disadvantages that the expression of one gene might have relative to another gene. I also suspect that the relationship between expression noise levels and mean fitness of the network is not biologically realistic; specific levels of noise for one gene with similar “outstrength” and “instrength” as another gene, may lead to different effects to the mean fitness between the two genes. Finally, as gene regulatory networks evolve, it does not seem that the immutable condition made for the regulatory interactions is valid. The following additional scenarios should be simulated and analyzed: 1) different selective pressures for different genes in a given nework, 2) network architectures with different levels of mean fitness from the evolved populations (see comment below), 3) in-silico evolution simulations for different populations sizes, a key parameter in evolution (not just N = 1000 (where computationally feasible); and 4) gene networks to evolving to change their regulatory interactions during the noise evolution step in the evolutionary simulation process.

Related to the comment above, the authors found that gene expression noise is reduced under a stabilizing selection regime, which is expected since the noise evolution simulations were performed for the gene network configurations with the highest fitness under stabilizing selection. Accordingly, since these gene networks achieved close to optimal fitness, any deviations from this due to gene expression noise is selected against and gene expression noise reduced. I would like to see simulations of scenarios where the evolution for gene networks that do not start close to the optimal fitness/scenarios where gene expression noise is beneficial (as is mentioned in the Introduction) and for different types of selection (i.e., disruptive selection and negative selection), where the outcomes are less intuitive. For the latter, at a minimum add a discussion on what you anticipate would occur if disruptive and negative selection were incorporated into your model.

Several important works on gene network architecture and the evolution of expression noise have not been mentioned/cited in this manuscript. Specifically, at a minimum, the following theoretical/computational and experimental studies from Mads Kaern, Gabor Balazsi, and Csaba Pal groups should be incorporated into the Introduction and Discussion sections of this manuscript: Charlebois, Phys. Rev. E, 2014 & 2015; Bodi, PLoS Biol., 2017; Camellato, Eng. Biol., 2019; and Farquhar, Nat. Commun., 2019.

Minor Concerns

Introduction: Define stabilizing selection for unfamiliar readers.

Methods & Results: I suggest removing the deterministic model from this manuscript, as it is confusing and seems out of place for a study focusing on gene expression noise. If it is essential to leave in the deterministic model, emphasize why this is so in the manuscript.

Results & Discussion: I suggest combining these sections to reduce repetiveness and to reduce the length of the manuscript. Alternatively, move all discussion of the results to the Discussion section.

Line 127: Based on the way the regulatory matrix is defined I assume that the genes in your model can autoregulate (though I did not see autoregulation in the figures)?

Lines 144-146: A justification and/or citation(s) should be provided for the assumption that there is no cooperative or competitive binding of transcription factors to transcription factor binding sites. Any limitations that result from making this assuption should also be discussed.

Lines 151-153: Is the using the normal distribution for gene expression justified? What about for bimodal gene expression that can occur for genes regulated by positive feedback gene networks (e.g., Nevozhay, PLoS Comput. Biol., 2012)? Same question for drawing the noise genotype mutations (Lines 201-202) and regulatory strength mutations (Lines 223-224) from a normal distirbution.

Lines 198-200: Gene expression in the absence of a selective agent can be costly (again see Nevozhay, PLoS Comput. Biol., 2012), yet it is assumed that there there is an equal constant value for the fitness of all phenotypes. This should be mentioned in the Discussion section of the manuscript.

Lines 227 – 228: Discuss the limitations of removing regulatory configurations that resulted in oscillating gene expression levels.

Lines 560-561: "GLMMs, however, make several assumptions that might be violated by the data in some cases. To further validate our conclusions, we computed the mutual information (MI) between variables." Explicitly state which assumptions are being referred to and why mutual information avoids these assumptions/provides more robustness.

**Have the authors made all data and (if applicable) computational code underlying the findings in their manuscript fully available?**

Reviewer #1: Yes

Reviewer #2: Yes

Reviewer #3: Yes

PLOS authors have the option to publish the peer review history of their article (what does this mean?). If published, this will include your full peer review and any attached files.

Reviewer #1: No

Reviewer #2: No

Reviewer #3: No
---

## [Decision Letter · Decision Letter 1]

27 Feb 2023

Dear Dr. Puzović,

We are pleased to inform you that your manuscript 'Being noisy in a crowd: differential selective pressure on gene expression noise in model gene regulatory networks' has been provisionally accepted for publication in PLOS Computational Biology.

Best regards,

Marija Cvijovic

Guest Editor

PLOS Computational Biology

Kiran Patil

Section Editor

PLOS Computational Biology

Reviewer's Responses to Questions

**Comments to the Authors:**

Reviewer #2: Thank you for responding thoroughly to my questions. Based on the authors' comprehensive answers, I believe this paper is suitable for publication.

Reviewer #3: Overall, my concerns have been addressed and support the publication of this manuscript in PLOS Computational Biology.

**Have the authors made all data and (if applicable) computational code underlying the findings in their manuscript fully available?**

Reviewer #2: Yes

Reviewer #3: Yes

PLOS authors have the option to publish the peer review history of their article (what does this mean?). If published, this will include your full peer review and any attached files.

Reviewer #2: No

Reviewer #3: No

---

## [Editor Report · Acceptance letter]

29 Mar 2023

PCOMPBIOL-D-22-01212R1 

Being noisy in a crowd: differential selective pressure on gene expression noise in model gene regulatory networks

Dear Dr Puzović,

I am pleased to inform you that your manuscript has been formally accepted for publication in PLOS Computational Biology. Your manuscript is now with our production department and you will be notified of the publication date in due course.

With kind regards,

Dorothy Lannert
